# "Mother in the making": Motherhood performativity of childless women in rural Pakistan

Rubeena Slamat[1,2]*, Piet Bracke[1], Melissa Ceuterick[1]

1 Department of Sociology, Faculty of Political & Social Sciences, Ghent University, Ghent, Belgium,
2 School of Management Sciences, Ghulam Ishaq Khan Institute (GIKI) Topi, Swabi, Pakistan

* Rubeena.Slamat@Ugent.be, rubeenasalamat@gmail.com

## Abstract

Motherhood is socially obligatory in rural Punjabi, Pakistan, leaving no room for voluntary childlessness. Women facing conception challenges strive to attain motherhood, combating the stigma of being labelled childless. Using Butler's performativity theory, this study delves into the experiences of childless women striving to become mothers in the pronatalist society of Pakistan. In-depth interviews with childless women and focus group discussions with community members were conducted. The data were analysed using a thematic analysis. Results revealed that women must demonstrate mothering qualities alongside seeking treatment to validate their desire for motherhood. The study concludes that societal discourses shape childless women's lives, influencing their conscious and unconscious adherence to gendered social norms of motherhood.

## Introduction

A child is an essential part of a woman's life in Pakistan [1–4]. Every couple, especially the woman, is expected to provide a child for the family. Childlessness is, therefore, considered unacceptable in the country. According to the United Nations Population Fund for Pakistan (UNFPA) [5, p. 110], 'voluntary childlessness does not seem to exist in the country'. Although childlessness is regarded as unacceptable, it does occur. Limited information is currently available about the precise prevalence of infertility and childlessness in Pakistan, and the existing data are notably outdated. The Family Planning Survey conducted by the National Institute of Population Studies (NIPS) in 2001 revealed that the infertility rate in Pakistan at that time was 21.9%, comprising 3.9% for primary infertility – women never conceived, and 18% for secondary infertility- women do not conceive after one year of unprotected sexual activity following the previous pregnancies [6]. In our study, childlessness is defined more broadly than infertility; it includes women who do not have living children due to primary infertility, secondary infertility, multiple miscarriages, multiple stillbirths, or neonatal deaths.

**Data availability statement:** The datasets cannot be shared publicly because they include the original information / identity markers of the participants. The participants did not agree to share the raw data outside of our team. We cannot share the data publicly for the participants' privacy. However, the data are available from the research group repository HEDERA for researchers who meet the criteria for access to confidential data. email ID: hedera.ugent@gmail.com.

**Funding:** The study is funded by the Higher Education Commission (HEC), Pakistan overseas grant no. 5-1/HRD/UESTPI(Batch-VI)/5464/2018/HEC and Flemish research council Fonds Wetenschappelijk Onderzoek (FWO) travel grant no. V407020N. The funders had no role in study design, data collection and analysis, decision to publish, or preparation of the manuscript.

**Competing interests:** The authors declared that they have no competing interests.

According to more recent figures from the UNFPA [5] and the Pakistan Demographic and Health Survey [7], 4.3% of married women of reproductive age (MWRA, 25–49 years) are childless in Pakistan. This survey did not include women who were separated, divorced, or widowed at the time of the survey. These statistics reflect the situation of women who are childless and living in a patriarchal and pronatalist society. Childless women are often seen as culprit and endure the physical and psychological consequences of childlessness [1,2,8–11].

Furthermore, childless women are often perceived as having a lower social position within the family and the community compared to women who do have children [1,2]. In Pakistani society, motherhood offers symbolic benefits in the form of respect and status. After giving birth, a woman is regarded as an important figure who participates in family matters and decision-making. Consequently, these women move upward in the social hierarchy of power, and especially in older age, they can regulate family matters [12].

Since Pakistani society equates womanhood with motherhood, childless women are supposed to achieve motherhood to complete their womanhood. Studies document that these women use various therapies to become a mother [2,13–19]. What these studies lack is an analysis of how these women's lifetime efforts to become a mother, and the impact this has on them. This study focusses on how childless Punjabi women are struggling to achieve motherhood and on the different acts they perform to show their desire to achieve motherhood. These women's live stories are the records of their life events, wishes, feelings, and performances which they mainly 'do' to avoid the stigma of childlessness [1,2]. To explain the experiences of these women we are using performativity theory developed by Judith Butler.

## Theoretical framework: Performativity theory

According to Butler, gender is a learned performance of gendered behaviour, what we commonly associate with femininity and masculinity. Gender is not something we are (or have within us) but rather something we continually do [20]. The central idea of this theory is that 'what is called gender identity is a performative accomplishment' [21 p. 463; emphasis added] involving recurring discursive imitation or 'recitation' of norms [20]. For example, individuals adopt specific style of dress, ways of speaking and moving according to their gender; as these choices are known as 'masculine' and 'feminine' and align with their sexed body. This constant repetition of norms maintains the illusion of gender as something stable [22]. So, gender identities are never complete but constructed and maintained in and through social interactions [20]. This idea is known as 'doing gender.' Butler argues that doing gender is not merely a performance of gender norms either deliberate or unconscious or habitual while performativity refers to the creation of a subject entirely in and through language, where 'the body becomes its gender through a series of acts which are renewed, revised, and consolidated through time' [23]. Butler gives three key aspects of the process of subject/gender identity formation: constitution, constraint, and subversion.

First, a constitution serves as a dominant discourse that establishes the rules for social order and includes methods for controlling and organising individuals (e.g.,

managing how people are positioned in space, monitoring them, and so on). These frameworks shape individuals by setting physical norms and regulating bodies. For instance, there are expected behaviours and ways of speaking for both men and women. Women, in particular, are taught to act and speak according to these norms. Social standards enforce these behaviours and often punish those who do not conform. For example, women who fail to meet beauty standards may face public criticism. Therefore, a person's gender identity is shaped entirely by social norms and discourse [21]. In other words, by continuously following and repeating these accepted norms, individuals become recognisable and understandable to both themselves and others. Similarly, societal discourses are available for 'motherhood' or good mothers [24–35]. These discourses demonstrate the do's and don'ts of the mothering role. Similarly, Pakistani society has set patterns for women to follow as mothers [36–41]. The adherence to these societal norms ensure that women align to their primary role as 'mothers'.

Second, discursive constraint refers to 'a set of repeated acts within a highly rigid regulatory frame' [20 p. 42] to maintain the gendered performances. These performances are shaped by a set of socially established meanings, known as the regulative discourses mentioned earlier. These discourses act like scripts that individuals are expected to follow, just as others have done before them. In this way, they naturally maintain identity. For this study, women are supposed to follow the norm of motherhood to maintain their identity. However, individuals are not entirely free to choose how they express their gendered selves [20]. The 'script' is already determined within this regulatory framework, and people can only choose from a limited set of gender styles [42 p. 56], much like selecting from a set of pre-approved 'costumes'. This idea is visible in this study where women are asked to provide a biological child to the family and society as the other women are/were doing.

As a result, the way individuals can express themselves is heavily influenced by restrictive gender norms, which are often rooted in heterosexual frameworks [43 p. 123]. Additionally, performing gender is a strategy for survival in a system [44] where deviation from these norms can lead to negative consequences, ranging from discrimination to violence. In Pakistani community, as mentioned above, motherhood is essential in women's lives. Women who cannot achieve this status bear the stigma [1,2,9]. Thus, 'doing gender' becomes a cultural necessity, making it difficult to step outside these roles [42]. While individuals are compelled to conform to be socially recognised, they still possess some agency. They can improvise within the script, but there are strict consequences for deviating too far from expected behaviour or making unauthorised changes to the performance [45]. The focus of the study is to explore childless women's performativity, specifically how they use their agency while conforming to their gendered identity.

Third, possibilities of subversion refer to the 'possibilities of deviance' from the set standards through discourse. The script is predetermined and provided, but social actors have some flexibility to improvise. According to Butler, the construction of gender identity is 'a temporal process which operates through the reiteration of norms; [it] is both produced and destabilised in the course of reiteration' [Butler, 21 p. 10] because it is impossible to perfectly replicate gender ideals. Individuals, therefore, have space for improvisation and variability in each performance [46]. Instead of being mere 'cultural dupes'‑passive victims of culture who simply absorb and reproduce social norms without agency or resistance, individuals can deviate from or go beyond the established norms [22]. It happens in some cases that individuals are unable to replicate these roles, as is the case with childless women in Punjab. Although, these women wish to be mothers they are unable to do so. (The different reasons for their childlessness are beyond the scope of this study and have been described previously by Slamat and colleagues [2]). These childless women have slightly different roles to play than mothers, as defined by the society [13,15]. The state of childlessness is seen as merely different and unwanted by society [1–3]. So, the developed societal discourses provide these persons chances of improvisation with restrictions of deviation. Childless women thus have some sort of agency or deviation possibility while living in a pronatalist society. Thus, the focus of this study is to document the lived experiences of childless women regarding the use of agency within societal constraints of a pronatalist society and thus how they are saved from becoming 'cultural dupes'.

Performativity theory, as explained by Butler, can be understood through the example of a wardrobe. Imagine a person who has a wardrobe full of different outfits. Some of these outfits may be beautiful, while others may be torn or worn out, but the person still has to choose one to wear. They cannot change the entire wardrobe. Similarly, the labels of 'masculine' and 'feminine' are assigned at birth. For instance, when a child is born and labelled a 'girl', this label defines not only her gender but also shapes many aspects of her life, including how she dresses, speaks, walks, and even eats. Society has pre-established norms and expectations for each of these behaviours, and the girl must navigate within these available frameworks or discourses as she grows up.

In summary, Butler [44] defines performativity as the interplay between sex, gender, and desire. She challenges the cultural belief that sex causes gender and that gender produces sex. In other words, deeply embedded cultural norms dictate standards of desire and contribute to the construction of gender identities. According to Butler, we recognise individuals not solely by their biological attributes but through the cultural roles they perform. Performativity involves a deep awareness of our actions, extending beyond stereotypical gender behaviours. Everything from clothing choices to speech patterns reflects the cultural performance of sex and gender, shaping our identities within society [44].

If motherhood is performative, it means that mothers are defined not only by biology but also by culturally shaped experiences. Ellen Ross [47] emphasises the importance of understanding mothers as performative 'subjects,' delving into their daily work, emotional bonds with children, and societal impacts. This broader perspective reveals that motherhood extends beyond biology, encompassing gendered cultural practices and societal institutions that reinforce a specific notion of female motherhood. Several studies document the ideals of motherhood and the performances of mothers accordingly (performativity theory).

According to Rich and McClatchy [48], the institution of motherhood is a socio-cultural construct and a version of patriarchal motherhood. This oppressive system is perpetuated by contrasting patriarchal discourses of 'angelic' mothering (the glorified, ideal, omnipotent, utopian mother) and 'monster' mothering (dominant, controlling, interfering, distant, over-protective, guilty, abusive, vicious) [49]. In their study 'Experiences of Motherhood: Challenging Ideals,' Frizelle and Hayes [50] explore how 'ideal' images of motherhood have oppressively persuaded mothers to strive for the unachievable in their everyday experiences, as they navigate the tensions between good and bad mothering.

Greenfield highlights the symbiotic relationship between the novel and mothering in *Mothering Daughters: Novels and the Politics of Family Romance: Frances Burney to Jane Austen* [51], considering motherhood as 'idealised and commonly represented as a full-time occupation' (p. 14), where 'women were defined by their maternity, and maternity was supposed to occupy a woman's perpetual interest' [pp. 14–15]. Similarly, Ruddick reminds us that the 'idealised figure of the Good Mother casts a long shadow over many actual mothers' lives' [52 p. 31]. Thus, community or shared discourses hold the power to regulate the lives of mothers, women, and individuals.

Sørensen [53] examines how working mothers navigate the expectations of 'good motherhood' alongside their professional responsibilities. Basnyat [54] investigates the dual identities of sex workers as both mothers and professionals. Similarly, as mentioned above Pakistani society has ideals of motherhood [55] and the women who cannot adhere to these ideals are seen as bad mothers [38,40,41,55]. This suggests that the ideal of motherhood influences the actions of many women, who strive to embody this ideal as a means of achieving a sense of accomplishment. Conversely, childless women actively seek motherhood, a desire evident in their daily actions and communication. Our study explores the experiences of childless women in rural Pakistani society, aiming to uncover the standard behaviours that express their desire for motherhood. This RQ is broken down in the following sub-RQs:

1. What are childless women in rural Punjab, Pakistan doing to achieve motherhood?

2. How do social norms regulate the life of these childless women to achieve motherhood?

3. How do childless Pakistani women use their agency on the journey to achieve motherhood?

 

## Research methodology

The village under study is located in central Punjab, between Lahore and Faisalabad. This neighbourhood is religiously diverse, with both Christian and Muslim communities. The village comprised 3,004 households [Record book of Basic Health Unit, 2020]. It has cemented streets, semi-paved roads that connect the village to nearby areas, and a semi-developed sewerage system. Villagers have access to electricity, basic education (schools), healthcare, and religious amenities (mosques and churches). As the two communities live together, the housing pattern is mixed.

The village community is patrilineal- the lineage is traced from male side or father side, patrilocal- where couple reside with husband's parents after marriage, and patriarchal- it is about the power of "men" over both women and other men, similar to other Pakistani communities [4]. The family is considered the smallest unit, headed by the husband as the breadwinner. The wife depends on him, and her main responsibility is homemaking [56]. Overall, the joint-extended family structure is preferred [4]. Marriage establishes a new family and sexual partnership; after marriage, the bride lives with her husband and in-laws. Motherhood is a woman's greatest source of power and pride in the village. Therefore, motherhood becomes a necessity for women [3], and childlessness is seen as a threat.

The villagers' primary means of subsistence is agriculture and farming, although some inhabitants work in government and private jobs. Men are mainly responsible for providing for their families, but sometimes women are also engaged in jobs.

## Data collection

Data for this study were collected by the primary author from January to June 2020. Ethical approval was granted by the Ethical Committee of the Faculty of Political and Social Sciences at Ghent University, Belgium under no. 2017−37. Permission to conduct research in the community was obtained from the relevant authorities through the Health Services Academy, Pakistan (the host institution). Participant recruitment began in the first week of January 2020 and concluded in the last week of March 2020. Target women, their partners, and older women were contacted through lady health workers (LHWs) [57], while the staff of the Basic Health Units (BHU) were approached directly.

The purpose of the study was explained to the research participants, and their consent was obtained—either in writing (for literate participants) or orally (for illiterate participants). A form was prepared for the informed consent including information of the researcher, institution's information, host institution's information, study participants' rights, i.e., they are willingly participating in the study, they have information regarding the research study, they can ask questions regarding it, they can quit it anytime and some more similar items. It also included the information regarding research study. I collected informed consent from all study participants written. The literate persons read and signed the consent.

Legal representative of the illiterate persons read and signed the consent on behalf of the participants (legal representative read the form, discuss it with 'to be participants', discuss with researcher and signed it).

A cover letter, prepared in consensus with all authors, was read aloud by the primary author to the participants prior to conducting interviews and focus group discussions (FGDs).

This article is based on 15 semi-structured, in-depth interviews with childless women (both primary and secondary infertility, as well as three women who conceived after many years of fertility treatments), and six FGDs involving 12 partners, four healers, and 16 older women. The interview participants (Table 1) were diverse.

All interviews and FGDs were conducted and moderated by the first author (RS), who is of Pakistani origin. This allowed her to explore themes in a culturally appropriate manner, as she could easily connect with the women. Additionally, as a mother of two, she was better able to address sensitive topics such as motherhood, sexual life, and health-related issues.

In Pakistani society, married women generally feel more comfortable discussing reproductive and sexual matters, especially with other women. Being a mother often allows them to engage in these discussions more openly, as their status

**Table 1. Participant' demographics.**

| Participants' Characteristics | Number |
|---|---|
| **Total women** | 15 |
| Primary infertile | 8 |
| Secondary infertile | 3 |
| Pregnant (after 9 years) | 1 |
| Mothers | 2 |
| **Age** | |
| 20-25 | 1 |
| 26-30 | 3 |
| 31-35 | 4 |
| 36-40 | 7 |
| **Length of marriage (in years)** | |
| 0-4 | 1 |
| 5-9 | 4 |
| 10-14 | 4 |
| 15-19 | 6 |
| 20+ | 0 |
| **Religion** | |
| Christian | 11 |
| Muslim | 4 |
| **Family type** | |
| Nuclear | 7 |
| Joint/extended joint | 8 |
| **Education** | |
| Illiterate | 7 |
| Primary | 0 |
| Middle | 5 |
| Matric | 3 |
| Higher secondary | 0 |
| **Husbands' profession** | |
| Labour | 5 |
| Shopkeeper | 2 |
| Farmer | 3 |
| Rikshaw driver | 2 |
| Office work | 3 |
| **Women's profession** | |
| Home maker | 14 |
| Private teacher | 1 |
| **Type of marriage** | |
| Endogamous | 6 |
| Exogamous | 9 |

as a mother provides a common ground. On the other hand, childless women may feel differently. They might share their emotions, experiences, and struggles more cautiously, believing that a researcher who is a mother may not fully understand their situation due to differing life experiences.

RS introduced herself several times as a researcher (not a health practitioner or gynaecologist), yet the women often perceived her as someone knowledgeable about health matters and sought her advice. The participants showed a strong interest in the study and provided detailed responses to the questions. All interviews were conducted in a private room within the participants' homes to minimise external pressures from family members. The interviews lasted between 35 and 60 minutes and were conducted either in the local language (Punjabi) or the national language (Urdu), based on the participants' preferences. RS is fluent in both languages.

All interviews and FGDs were recorded after obtaining consent from the participants. RS transcribed the interviews and FGDs in the local/national language, and then translated them fully into English [58]. Key terms and proverbs were transcribed following the methodology of Al-Amer, Ramjan, Glew, Darwish, and Salamonson [58]. Two language experts—proficient in both the local/national language and English—rechecked all transcriptions, also in line with Al-Amer, Ramjan, Glew, Darwish, and Salamonson [58].

In order to maintain the confidentiality of participants, all interviews were anonymised prior to being submitted for language review. Each interview was assigned a unique code, and all personal identifiers, including names, ages, and marital statuses, were removed. This coding system was consistently applied throughout the data analysis process to ensure the privacy of participants was protected. The use of codes was a key measure to safeguard the anonymity and confidentiality of the participants involved in the study.

Furthermore, this study focuses on an underprivileged segment of society, namely childless women. The researcher empathised deeply with their challenges and sought to give voice to their experiences through this research. During the analysis process, numerous accounts of hardship and struggle emerged, reflecting the difficulties these women face. The researcher made a conscious effort to document their experiences faithfully, neither exaggerating nor minimising their struggles. To ensure the authenticity of the representation, the researcher remained attentive to the women's own narratives, prioritising their expressions and perspectives throughout the study.

## Data analysis

The data were inductively analysed using thematic analysis, following the steps outlined by Braun and Clarke [59], and conducted by the first author. Initially, the interviews and FGD excerpts were read iteratively to facilitate familiarisation. In the second stage, the data were coded by assigning initial codes, starting with an interview (coding based on the information in the excerpts). After coding five to six interviews, the codes were consolidated to create a coding tree. All authors evaluated these codes, which resulted in a final coding tree. The remaining data were then coded in accordance with the established coding tree.

In the third stage, similar codes were grouped based on shared information. For example, codes related to treatment practices were grouped together under an overarching code, which helped identify emerging themes. During the fourth stage, the themes were refined. Some themes were merged to construct a coherent narrative, while others, which lacked coherence, were broken down into codes again to ensure better thematic alignment or to form new themes. For instance, codes related to 'women's bodies perceived as problematic' and 'the behaviour of family members towards women during certain occasions' appeared to overlap. These codes were restructured and merged into new themes: 'women following treatments' and 'gender dynamics'.

In the fifth step, the themes were finalised and named. At this stage, the major themes of the article were agreed upon by all authors. Finally, these themes were organised coherently to structure the article, with their alignment contributing to the overall story flow.

## Findings and discussion

### The discourses (constitution) of marriage pointing towards motherhood

A first theme that we discerned in our dataset is the dominant discourse that equates marriage with the expectation of motherhood. This theme illustrates the first point of performativity theory: that cultural discourses serve as – in Butler's

terms- constitution and community members are socialised to follow these. Community members state that a woman is supposed to become a mother as soon as possible after marriage, as was stated in a FGD:

> *"A girl becomes a woman after becoming a mother."* (FGD1, participant 2)

The excerpt shows that marriage is considered an entry point to motherhood. A woman entering a marital relationship has to take the responsibility to become a mother. By doing so, she is performing womanhood, in line with the 'constitution.' This responsibility is also illustrated in other discursive forms like prayers and jokes. Participants stated that the prayers given to women during a marriage ceremony are all related to motherhood. Examples of prayers given are:

> *"God bestow you children; god bestow you a son."* (FGD 3, participant 5)

The excerpt shows that the prayers given by older women remind a young woman and inscribe her to become a mother from this point onward in her life. The ceremonies and rituals with repeated prayers of motherhood together form the 'constitution' that assign her to achieve the status of a mother. Moreover, during these ceremonies, emphasis is placed on other women who already hold a significant social standing. Specifically, married women cohabiting with their spouses who are already raising healthy children are invited to participate in the performance of these ritual, as stated by this woman:

> *"My aunt called me to apply henna at my cousin's hand during her mehndi (ceremony)…because I have two sons."* (Interviewee 6)

In Pakistani cultural traditions, certain rituals are performed upon a bride's arrival at her in-laws' home, signifying her integration into the new family and societal expectations. One such ritual involves placing a young (male) child, typically between the ages of three months to two years, in the bride's lap. This ritual symbolically reinforces the expectation that the bride will soon take on the role of motherhood. The ritual, often performed on the bride's first day at her in-laws' home, serves not only as a welcoming gesture but also as a subtle imposition of the societal role of a mother, reflecting the cultural emphasis on fertility and procreation. These ceremonies, rituals, and activities prompt the bride to reflect on her future responsibilities. As she transitions from girlhood to married womanhood, she is expected to fulfill the duties and obligations tied to her new family role, particularly those concerning her in-laws.

Similarly, the woman's friends, either married or unmarried, make jokes relevant to her life after marriage.

> *"Next year, you will have a shining son or a fairy-like daughter."* (Interviewee 8)

The above-mentioned excerpt shows that around the time of marriage, jokes, talks, and communication all revolves around marriage and having children. Friends of the woman expect her to have a child in the coming years and married friends share their experiences in the shift of their relationship after having a child. These friends expressed that after becoming mothers, their husbands became more caring and affectionate towards them. This shift in dynamics made them feel empowered. Participants use several words and proverbs while discussing this theme are: *chand sa beta* (beautiful son)*, pari* (beautiful daughter)*, maan hona aik ezaz he* (motherhood is an honour)*, bacha zanjeer he jo mann baap ko bandhta he* (child is a chain that joins parents)*, maan banna her ourat ka haq he* (a woman has a right to become a mother)*, maan ke peron ke neeche jannat he* (mother has heaven under her feet)*, khana peena to maan behnen bhi deti hen per oulaad sirf biwi deti he* (mother and sister of a man can fulfil other duties but wife can give him the children).

Hence, marriage is regarded as the initial step or initiation into motherhood, as societal expectations dictate that every married woman is expected to bear children to continue the family and lineage. Consequently, each married woman

assumes the responsibility of perpetuating the family line by ensuring the birth of a child [1–4]. Attaining the status of motherhood is thus seen as an obligatory task for women [1–4], with the community insisting on the fulfilment of this task shortly after marriage.

**The bride in limbo**

This theme focuses on the transitional state of women when they are neither bride nor mother. This theme has a connection with the first point of performativity theory: means cultural discourses serve as 'constitution' where individuals are trained and scrutinised in a strict manner according to the cultural discourses. It highlights how family and community members begin questioning the fertility and social status of women shortly after marriage. Participants stated that after marriage, the most anticipated news is the bride's pregnancy. As a result, neighbours often inquire about pregnancy, directing questions to the mother-in-law, married sisters-in-law, or the bride herself. This point was discussed in a FGD:

> *"Relatives (women) ask the mother-in-law, 'Is there any good news?'"* (FGD 2, Participant 1)

These excerpt illustrate the community's intense curiosity about a newlywed bride's potential pregnancy. Sometimes, they even interpret the bride's visit outside the in-laws' home as a possible trip to the hospital for pregnancy-related reasons.
Another participant shared:

> *"I went to my sister's home for lunch (she arranged it for the new couple)…When I came back, the elderly neighbour asked, 'Are you coming from the hospital?'"* (Interviewee 2)

The community shows a vested interest in the fertility of the new bride. Many older women compare the pregnancies of different newlywed couples, drawing attention to those who have not yet conceived. One woman shared:

> *"After six years of marriage, I was still not pregnant….Women who got married around the same time as me, or even after, all have children now…but I don't."* (Interviewee 5)

This excerpts reveals that community members often compare women based on pregnancy status, number of children, and the time between marriage and first pregnancy. When a woman does not conceive within two or three years, suspicions arise, casting doubt on her fertility. After five to six years, such women are labelled as 'childless' and are seen as needing medical intervention to conceive. This underscores the societal expectation for women to transition quickly from bride to mother, rather than remaining in a liminal phase.
Participants used various phrases when discussing this theme: *kuch he* (are you pregnant?), *koi khabar* (do you have news, implying pregnancy), *tu wi koch ker le* (you should do something to become pregnant), *tere wass di gaal nae* (pregnancy is not your concern), *rab kher kare, waya noun 5−4 saal ho gaye ne* (God bless her; she's been married for about five years), *koi masla wi ho sakda ae* (she may have an issue, possibly infertility), *saas noun tayaan kerna chahi de* (her mother-in-law should focus on this matter).
Since these women do not fulfil the role of 'motherhood' as expected, and thus deviate from gendered norms, they face potential stigmatisation, as discussed extensively elsewhere [1,2]. This theme reflects that the community typically expects a child from a newly married woman within one year of marriage. A 'waiting period' of up to two years is accepted without suspicion, during which a woman is expected to give birth or at least announce a pregnancy. After this, the community's suspicions about her fertility increase with each passing year. Over time, the woman is seen as someone who has failed to fulfil her expected role, intensifying the stigma she faces [1,2].

## Treatment seeking as an expression to achieve motherhood

This theme explores women's experiences with health-seeking and communication surrounding treatments in their journey to become mothers, specifically those who cannot conceive within what society considers the 'right time' — typically a few years after marriage. This theme relates to the second point of performativity theory, 'discursive constraint', in which women replicate the behaviours. The women make efforts to provide a 'biological child' as expected by society. These women, in attempting to meet gender expectations, seek out and follow various treatments.

A woman stated that:

*"I have visited many doctors, but every doctor told me a different (medical) problem. I took medicine from each doctor and used it."* (Interviewee 10)

Another woman shared her story:

*"I contacted religious leaders. I took amulets. If someone recommended a hakeem (herbalist), I would visit him for treatment. [...] I consulted all the gynaecologists practicing in the nearby city… you know, I had hope while following each treatment… but I wept every time at the end when there was no news of pregnancy."* (Interviewee 8)

The above-mentioned excerpts shows that childless women were following the treatments ranging from home remedies to bio-medical medication. In seeking treatment, they turn to a variety of options, including home remedies, religious practices, herbal treatments, and modern medicine. Studies indicate that women use herbal decoctions, apply pastes to their bodies, and receive massages [16–19]. They also consume charmed water, jaggery, black pepper, and visit religious places, such as shrines, for prayers [60,61]. These treatments are often repeated multiple times over the years. Thus, these women start following treatments soon after marriage and repeatedly shift among various healing categories and practitioners. Their accounts show a wide array of treatments pursued in their efforts to become mothers similar to other regions [13,15,62–67]. Furthermore, treatment-seeking is a journey fraught with emotion and grief. Each treatment begins with hope, but every unsuccessful outcome brings sorrow. As women, they feel compelled to continue, pressured by societal norms not to abandon their pursuit of motherhood. The repetition of treatments following is a way of motherhood performativity and complying the norms of 'mother in making'.

Women also shared that their male partners are not subjected to the same expectations for medical consultations. Typically, male partners only seek medical advice if a gynaecologist explicitly requests it. One woman recounted:

*"The doctor asked for my husband's lab reports... I took his report to the doctor, who said, '...your husband has weak sperm.'"* (Interviewee 1)

This excerpt shows that although male partners may contribute to infertility, they often seek treatment only upon a doctor's recommendation. Community expects women to prioritise proving their fertility. By doing so, women are following the gendered scripted norms of society. Participants also expressed that even if their husbands have fertility issues, women are still encouraged to seek treatments and consult various healers, either traditional or professional. Consequently, these women may also face suspicion regarding their fertility. As one woman explained:

*"My mother-in-law send both of us to the doctor…..My husband is taking medicine, but I go to the doctor with him."* (Interviewee 4)

Thus, women are recurrently expected to demonstrate their physical fitness, fulfilling their prescribed gender role of a 'mother in the making.'

Furthermore, women have to follow the cultural discourses while discussing the cause of childlessness and physical fitness. It is the cultural expectation that a woman should elaborate regarding her physical fitness and treatment cycles daily and occasionally. At the same time, she is socially sanctioned to does not discuss her husband's infertility as a cause of childlessness. This announcement is seen as inappropriate culturally. A woman shared:

*"Sister… it's a daily routine… I tell other women about each treatment, from the first to the last… I feel sad every time… but people don't understand; they just suggest another treatment."* (Interviewee 8)

Another woman shared:

*"My husband has an issue of low sperms (low sperm count)…..I am physically fit. But we at home know that he has this problem."* (Interviewee 4)

The excerpt above indicates that women assure community members of their physical fitness and future potential to bear a biological child. Consequently, women often find themselves emphasising their well-being by repeatedly affirming the practitioner's statement that they are in good physical condition and have no medical issues. However, participants also expressed that they are not permitted to discuss their husbands' physical condition, especially when the husband has been diagnosed with infertility issues [61,68–70]. Cultural norms restrict childless women and demand a different type of behaviour from them. In such cases, the woman is expected to stay silent about her husband's infertility and to endure the pain when, in public gatherings, community members discuss her childlessness. By protecting their husbands and bearing the full burden of childlessness, these women are conforming to prescribed cultural expectations.
Another participant stated:

*"Every time someone asks about my pregnancy… I feel heartbroken… I start crying… it happens automatically… but I notice people show sympathy right after I cry."* (Interviewee 9).

Above-mentioned excerpt shows that women express grief openly, crying when discussing their childlessness, which is perceived by the community as a very explicit form of performing the desired role (by woman). Feelings of grief [71] are seen (in the community) having a desire for motherhood. Community members, in response, suggest that they seek more healers and continue treatment. Thus, women bear the burden of childlessness [1,2,69,72,73] to comply the scripted role of motherhood.
Furthermore, expressions commonly used in this context, such as *Allah kher kare ga* (God will do better for you), *Allah kabhi toh sunay ga* (God will hear you one day), and *Harakat mein barkat hai* (There is blessing in effort), encourage women to keep searching for more treatments. These phrases reinforce the expectation for women to continue their journey, seeking solutions tirelessly.

**Love for other's children as a motherly characteristic**

This theme has a connection with the second point of performativity theory-'discursive constraint'. It discusses the expectation that women must display *mamta* (motherly instinct) and affection for children. By doing so, even childless women can fulfill the expected gender role by showing affection for others' children. Community members shared that a woman's behaviour toward other children suggests a desire to become a mother. One woman explained:

*"I love children very much…I usually take care of my husband's brothers' children...the older women say to me, 'You love children so much…God will bless you with a child.'"* (Interviewee 12)

Another woman state:

> *"My husband has three brothers, all married, so we have many children at home…I stay busy with them…I care for them when their mothers go to visit neighbours or the market…sometimes I feel tired, but I never refuse…they might feel hurt."* (Interviewee 3)

These excerpts illustrate that when women display nurturing behaviour, it is socially appreciated as an expression of their desire for motherhood. Conversely, any perceived lack of affection is met with criticism, with childless women being labelled as 'unfit mothers.' Negative reactions often provoke painful taunts, such as *tum maa banne ke qabil nahi ho* (you are not capable of becoming a mother). Such remarks emotionally burden these women, who respond by actively portraying a nurturing image, conforming to the societal expectation of a 'mother in the making.'

As noted, childless women are expected to show constant affection toward other children. This demand often encroaches upon their personal lives. For example, when family members live in the same compound, they frequently leave their children with these women while they go out (to visit neighbours, relatives or the market), assuming they will take care of the children. These women fulfill this role by feeding, storytelling, playing and attending to the children's needs. Participants shared that they often sacrifice their rest time to care for their siblings' children. Thus, showing love and care for children is regarded as an essential aspect of motherhood, and by doing so, these women comply with this societal expectation.

## Adopting motherhood

This theme is connected with the third point of the performativity theory 'subversion'. It discusses the solution of childlessness through adoption. Society offers adoption as an option for childless couples who have tried extensively to conceive but remain unsuccessful. Participants shared that some couples choose to adopt children after many years of marriage. One woman stated:

> *"I should adopt a child; the idea came to me suddenly. I discussed it with my husband... he agreed... and then I adopted my sister-in-law's child."* (Interviewee 4)

> *"Sister! I think about adoption. But my husband does not agree... I have tried many times to persuade him, but he has refused."* (Interviewee 6)

These excerpts illustrate that adoption is a viable option within the community. Some childless women in the community have chosen to adopt, consistent with findings in previous studies [8,13,14,74–77]. Women who adopted children from their own siblings or relatives are enacting motherhood in a way that diverges from traditional norms, as they do not replicate the scripted role of a biological mother. This variation in the practice of motherhood suggests that culture permits some flexibility from established norms. However, individuals are still constrained from freely deviating from these norms.

In this community, cultural discourses strongly influence deviations from traditional norms, especially regarding childless women considering adoption. Following specific guidelines—such as adopting a child of the same caste and religion, knowing the biological parents and preferring familial ties—is essential. A woman expressed her wish as follows:

> *"Sister, I wish to adopt a child but you know we could not find a child in our siblings for adoption. We could not adopt a child from outside our family." (Interviewee 13)*

Obtaining consent from the husband and parents-in-law is also critical for adoption. Some women refrain from adopting due to factors like family approval and the availability of children. In some regions adoption is common [78,79] and these

nations such as US, India and South Africa impose strict rules on couples who adopt children [78–81]. India has launched a digital system with record of the children that could be adopted [82]. In US the adoption is done through foster houses [78]. Unlike these regions where adoption is more common [78–80], this community imposes strict standards on adoption for childless women. It is seen as a family matter. Similarly, in Bangladesh, women wish to adopt a child but they are supposed to follow the decisions of their husband and families [67,73]. Nahar documented that adoption is unpopular in South Asian countries [67].

The community employs specific terms to discuss this theme, indicating the subversion of traditional scripts by women. These words reflect a (accepted) deviation from social norms. Examples include *saggi maan nai* (not a biological mother), *lepalak*/*goed laya*/*mangya* (adopted child), and *saggi oulad nahi* (not a biological child). Additionally, discussions often reference the relationship between the child and woman, such as *bhateeja*/*bhanja* (son of her brother/sister) and *jeth*/*deyoor*/ *nannaan da munda*/*kuri* (son/daughter of her husband's brother/sister).

The use of terminology in this theme shows that the adopted child is never considered the own child of the woman, rather, the adopted child, to some extent, satisfy her mothering feelings. Also, the woman behaviour towards that child shows the community that she has mothering qualities. The woman through her caring behaviour, bringing up a child and remain busy with child whole day explicitly fulfil her role of a mother as expected by the society. Although, the woman cannot provide a biological child but her connection to the adopted child gives her a footing in the family and society. Mostly, the adopted children belongs to husband's lineage, so in this way she maintains a connection with the in-laws family.

## Conclusion

This study explored the performative enactment of motherhood by childless women in rural Punjabi-Pakistan. In Pakistan, motherhood is considered a social mandate. Community discourses at the time of marriage remind women of the necessity of achieving a maternal identity by giving birth. However, even childless women sustain an identity of being 'mothers in the making' by embodying behaviours, communication styles, and acts deeply inscribed in the culture. The study findings show how these women make efforts to become mothers, avoid the label of childlessness, and become part of streamlined women group 'mothers'. Their daily efforts, selfless love for their siblings' children, and patience are ways to be recognised as appropriate (and potential) mothers according to ideal societal motherhood discourse.

For years, childless women often pursue treatments, share stories of their medical experiences, and demonstrate their physical readiness to fulfill the culturally gendered expectations of motherhood. In the contemporary patriarchal society of Pakistan, the burden and responsibility to seek treatment and to provide a healthy child to the family, is placed on women. Many of these women express a desire for motherhood by displaying maternal qualities toward the children of their siblings and their husbands' siblings. These repeated behavioural patterns show that women are 'replicating the behaviours in a regulatory frame' as previous women have done, as an expression that they are performing their 'expected role'. Cultural discourses—such as scrutiny for signs of possible pregnancy, encouragement to seek treatment, and expressions of sympathy—keep these women firmly within a maternal role. This pressure pushes them to do the repeatedly same acts to uphold an identity as 'mothers in the making.'

Furthermore, some women exercise agency by choosing to adopt a child, shaping new life plans around this decision. They actively perform their maternal identity by discussing their feelings and experiences related to the adopted child. The adoption is a form of 'subversion' but it is also controlled by the cultural discourses. For instance, the husband and in-laws are important figures whose agreement is necessary for adoption. Additionally, there are certain requirements regarding the ideal child 'to be adopted' that should be fulfilled. Thus, adoption itself is also shaped by cultural expectations, with these women maintaining an identity by 'praying for a biological child' and voicing both their hopes for adoption and the potential obstacles posed by their husbands' and in-laws' families. The behaviours surrounding the 'adoption' shows that cultural discourses have a strong control on ways of subversion.

In conclusion, this study suggests that individuals are both consciously and unconsciously shaped and directed by cultural discourses to perform certain roles in daily life. Their actions follow social regulations aligned with their community's expectations such as childless women in this study. While perfect replication of cultural norms is impossible, culture allows for limited, accepted deviations where replication is not feasible. Thus, our identities are ultimately determined by how we exist in a culture [44].

Through this study, we urge policymakers to develop policies for the betterment of these women. First, an updated national survey on infertility and childlessness could help identify the scale of the issue, inform the need for expanding infertility clinics across the country, and promote positive attitudes through public campaigns. The health sector should also ensure the availability of affordable, well-functioning infertility clinics and raise awareness about infertility-related issues. Furthermore, more gender-sensitive research is needed on the causes of infertility in both women and men in these regions.

The novelty of the study is that it explores the experiences of childless women and their struggles to show their efforts to fulfil the expected role of 'motherhood' in the patriarchal and pronatalist society of Pakistan.

Our study has some limitations. First, it focuses exclusively on women. A separate study should be conducted to document the experiences of childless men as a 'father' in the patriarchal society of Pakistan. Furthermore, the study setting was limited to a particular rural village, with largely similar socio-economic circumstances. Therefore, the impact of socioeconomic status (related to geographical location) on treatment choices has not been explored.

Despite these limitations, this study is contributing to the body of literature on motherhood and childlessness by documenting the lived experiences of under-privileged women in a pronatalist nation.

## Acknowledgments

The authors are grateful to all participants for sharing their stories and views. We also thanks to all persons and institutions who facilitated us to conduct this research study.

## Author contributions

**Conceptualization:** Rubeena Slamat, Piet Bracke, Melissa Ceuterick.

**Data curation:** Rubeena Slamat.

**Formal analysis:** Rubeena Slamat.

**Funding acquisition:** Rubeena Slamat.

**Investigation:** Rubeena Slamat.

**Methodology:** Rubeena Slamat, Melissa Ceuterick.

**Project administration:** Piet Bracke.

**Resources:** Piet Bracke.

**Software:** Rubeena Slamat.

**Supervision:** Piet Bracke, Melissa Ceuterick.

**Validation:** Piet Bracke, Melissa Ceuterick.

**Visualization:** Rubeena Slamat, Piet Bracke.

**Writing – original draft:** Rubeena Slamat.

**Writing – review & editing:** Melissa Ceuterick.

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
