## [Decision Letter · Decision Letter 0]

15 Jul 2024

Dear Dr. Slamat,

Thank you for submitting your manuscript to PLOS ONE. After careful consideration, we feel that it has merit but does not fully meet PLOS ONE’s publication criteria as it currently stands. Therefore, we invite you to submit a revised version of the manuscript that addresses the points raised during the review process.

Please note that we have only been able to secure a single reviewer to assess your manuscript. We are issuing a decision on your manuscript at this point to prevent further delays in the evaluation of your manuscript. Please be aware that the editor who handles your revised manuscript might find it necessary to invite additional reviewers to assess this work once the revised manuscript is submitted. However, we will aim to proceed on the basis of this single review if possible.

We look forward to receiving your revised manuscript.

Kind regards,

Vanessa Carels

Staff Editor

PLOS ONE

Journal Requirements:

"Higher Education Commission (HEC) Pakistan, Overseas grant

Wetenschappelijk Onderzoek – Vlaanderen (FWO) Travel grant"

5. In this instance it seems there may be acceptable restrictions in place that prevent the public sharing of your minimal data. However, in line with our goal of ensuring long-term data availability to all interested researchers, PLOS’ Data Policy states that authors cannot be the sole named individuals responsible for ensuring data access (http://journals.plos.org/plosone/s/data-availability#loc-acceptable-data-sharing-methods).

Reviewers' comments:

Reviewer's Responses to Questions

**Comments to the Author**

1. Is the manuscript technically sound, and do the data support the conclusions?

Reviewer #1: Yes

2. Has the statistical analysis been performed appropriately and rigorously?

Reviewer #1: N/A

3. Have the authors made all data underlying the findings in their manuscript fully available?

Reviewer #1: Yes

4. Is the manuscript presented in an intelligible fashion and written in standard English?

Reviewer #1: No

Reviewer #1: 1. Summary of the Research and Personal Overall Impression

The research brings out issues that trouble women failing to have children. Motherhood performativity of childless women is evident in rural Punjab. The research reveals that they are seen as incomplete because of childlessness. When they get children (biologically or by adoption) and love them, they will have performed what is socially required for their gender. This study revealed that they are seen as responsible for childless marriages. Men are excused from impotence.

I commend the use of Butler’s Performativity theory as it amplifies the lived experience of involuntarily childless women. The researchers managed to highlight areas that a women must perform to validate her gender.

What I like about this research is the choice of a rural context for the study of involuntary childlessness. This is where the cultural expectations are undiluted by modernisation. The context reveals what a specific people group believe before their culture is influenced by other cultures.

2. Discussion of Specific Areas for Improvement

The Conceptual Framework

Instead of describing Butler’s Performativity theory in depth, the researchers were supposed to highlight the main key factors and how they are related. In this case, women’s identity or gender in rural Punjab is validated by their performativity. Bearing children in this study is the main “performance” that women can use to validate their gender.

Research Methodology

The research methodology section on page 8 and 9 is discussing the research context.

The researchers didn’t highlight the limitations of their study. For instance, the primary researcher was a mother of two children. What effect did that have on the data analysis? Did the self of the researcher have any impact on the presentation of findings? If not, what measures were put in place? Or how was that achieved?

Data Analysis

In data analysis, the researchers need to reduce the number of direct quotations of the participants. They can paraphrase them and give their interpretation of the excerpts. In some places, there is very little data analysis but more participant quotations. Readers will need to hear what the researchers make of the collected data.

How is the cited excerpt speaking to motherhood performativity of Punjabi childless women? How does it address the implications of the exhibited behaviours and further reveal the importance attached to motherhood? How does the gendered expectation of mothering a child regulate the lives of these childless women as they work towards becoming mothers?

Flow of Thought

The clarity of some of the research findings is clouded by language articulation. While there are some minor typographical errors, there are some portions in the manuscript that will need expert language editing. This will enhance the readability of the article.

The researchers need to allow an interaction of the authors they cite. They can employ using joining words to make the transition from one author to the other smoother. On page 5 and 6, the authors have liberally used direct quotations. They can paraphrase the quotes to avoid monotony.

A minor observation is on use of respondents and participants interchangeably throughout the paper. This article has participants rather than respondents. The target population provided qualitative data.

The section which discusses the bride in limbo (page 15 onwards) is not very clear. The connection between the subtheme (mentioned in the title) and the body needs to be strengthened. The researchers have a good intention; however, more clarity is needed to openly bring out the finding on motherhood performativity in Punjabi childless women.

Confidentiality

In the data collection paragraph (page 11, last line), I am concerned about confidentiality issues. If “All transcriptions were rechecked by two language experts,” what did the researchers put in place to protect the participant’s identity? Clarity on that may help

In data analysis, when coding the participants, mentioning their age may reveal their identity to those who may have seen the participants. I suggest a revised coding system.

Layout of Manuscript

Page numbers are missing. This is a minor issue that can be rectified.

References and In-text Citations

On page 27, reference number 7, the researchers must follow the guidelines for published media (print or online newspapers and magazine articles).

The numbering of references did not follow the order they appear in the text. Authors should refer to the author guidelines for this.

I have a minor observation on intext citations throughout the paper. According to the guidelines, in the text, the reference number must be cited in square brackets. The researchers used parentheses.

The authors must align all other references to the Vancouver reference style.

Recommendation

I recommend the article for publication. It presents a crucial matter that needs to be discussed. However, the above revisions can be taken into consideration to strengthen the article. More attention should be given to data analysis. In some places (especially the first section on “The constitution of marriage pointing towards motherhood”, over-quoting the participants repressed in-depth analysis.

**Do you want your identity to be public for this peer review?** For information about this choice, including consent withdrawal, please see our Privacy Policy

Reviewer #1: **Yes:** Sikhumbuzo Dube

---

## [Author Response · Author response to Decision Letter 1]

21 Jan 2025

Respected editor,

We are thankful to the editor and reviewer for their effort and time. We have incorporated all suggestion by the editor and reviewer. A point by point response is uploaded in a word file.

Thanks again,

Kind regards,

Rubeena Slamat

---

## [Decision Letter · Decision Letter 1]

24 Jul 2025

Dear Dr. Slamat,

Thank you for submitting your manuscript to PLOS ONE. After careful consideration, we feel that it has merit but does not fully meet PLOS ONE’s publication criteria as it currently stands. Therefore, we invite you to submit a revised version of the manuscript that addresses the points raised during the review process.

We look forward to receiving your revised manuscript.

Kind regards,

Jenna Scaramanga

Staff Editor

PLOS ONE

Journal Requirements:

Additional Editor Comments:

 Could you please revise the manuscript to carefully address the concerns raised? Please note that Reviewer 3’s comments appear largely in an attachment. 

Reviewers' comments:

Reviewer's Responses to Questions

**Comments to the Author**

Reviewer #2: All comments have been addressed

Reviewer #3: (No Response)

Reviewer #4: (No Response)

2. Is the manuscript technically sound, and do the data support the conclusions?

Reviewer #2: Yes

Reviewer #3: (No Response)

Reviewer #4: Yes

3. Has the statistical analysis been performed appropriately and rigorously?

Reviewer #2: N/A

Reviewer #3: (No Response)

Reviewer #4: (No Response)

4. Have the authors made all data underlying the findings in their manuscript fully available?

Reviewer #2: No

Reviewer #3: (No Response)

Reviewer #4: (No Response)

5. Is the manuscript presented in an intelligible fashion and written in standard English?

Reviewer #2: Yes

Reviewer #3: (No Response)

Reviewer #4: Yes

Reviewer #2: This manuscript presents a valuable contribution to our understanding of how childless women in rural Pakistan navigate and perform their gendered identities within a cultural context that strongly equates womanhood with motherhood. Using Butler's performativity theory as a theoretical framework, the authors effectively demonstrate how these women maintain an identity as "mothers in the making" through various performances that align with cultural expectations while exercising limited forms of agency.

Minor Suggestions

1. Theoretical Applications:

While the theoretical framework is well-established, there could be more explicit connections between Butler's concept of "subversion" and the adoption practices described in the fifth theme. This would strengthen the analysis of how women exercise agency within constrained cultural contexts.

2. Comparative Context:

The manuscript could benefit from brief comparisons with findings from studies in other cultural contexts to highlight what might be unique about the Pakistani rural context versus what might be more universal experiences of childless women in pronatalist societies.

3. Policy Implications:

Consider adding a brief discussion about the potential implications of these findings for healthcare providers, community workers, or policymakers working with childless women in rural Pakistan or similar contexts.

4. Intersectionality:

Although the paper acknowledges differences in religion, education, and marriage duration among participants, a more explicit discussion of how these factors might intersect to shape women's experiences of childlessness would enhance the analysis.

Overall, this is a well-executed qualitative study that makes a significant contribution to understanding the gendered experiences of childless women in rural Pakistan. The manuscript effectively demonstrates how these women navigate cultural expectations through performances that affirm their maternal identity and desire, even in the absence of biological children. The paper provides valuable insights into the complex interplay between cultural discourses, gender norms, and individual agency in shaping women's lives and identities.

Reviewer #3: (No Response)

Reviewer #4: This research is novel and innovative. The methodology is sound and well-described. However, there are some issues which need addressing in order to improve the quality of the manuscript. Please see my recommendations below.

The manuscript would benefit from one more professional edit. There are some small expression issues that need to be ironed out.

There are several points in the manuscript where the research is referred to as ‘my study’ even though there is more than one author listed. Also, in the analysis section, it is revealed that the first author conducted the analysis by themselves. Please correct me if I am wrong about this. It is unclear to me what the contribution of the other authors consists of. It should be more evident in the manuscript how all contributed.

The abstract highlights the main contribution of the article as showing how ‘societal discourses shape women’s lives.’ However, this is not necessarily new in the literature. What is the study adding to what we know about motherhood, performing gender, involuntary childlessness? This should be better emphasised in the abstract, introduction, literature review and conclusion.

Related to the point above, there is too much dwelling on performativity and explaining Butler in the literature review. There should be more connections made with the broad literature on gender performativity, motherhood expectations, and infertility. What do other studies say about these topics and how can this research contribute to these conversations even if they are not necessarily located in Pakistan.

I am wondering about the rationale behind broadening the childless category. Is it because all participants experienced the pressures of motherhood at some point? This should be specified early in the manuscript.

Below table 1 there are two paragraphs that are an exact copy of each other.

Theme titles are long and convoluted. I recommend shortening them to make the key message of each theme clearer.

The conclusion needs to be strengthened. It does not discuss any of the broader implications of the findings. This is a shame because the analysis is excellent. I already recommended that more relevant literature should be added to the review in the first part of the manuscript. The conclusion should connect to this literature (on motherhood, childlessness, etc.) by telling us how the findings contribute to scholarly work that is related, but not specific to Pakistan. There is an opportunity to strengthen the relevance of your findings, and this should not be missed.

Best of luck with revisions.

**Do you want your identity to be public for this peer review?** For information about this choice, including consent withdrawal, please see our Privacy Policy

Reviewer #2: **Yes:** Shinnosuke Komiya

Reviewer #3: No

Reviewer #4: No

---

## [Author Response · Author response to Decision Letter 2]

5 Oct 2025

Respected reviewers,

We are thankful to the reviewers of this paper for their interest and insightful suggestions; we appreciate their effort and valuable suggestions. We have taken all the reviewers' comments on board and tried our best to incorporate them. The responses to the comments are given in a document response to reviewers. The file is uploaded in the attachment section.

Kind regards,

Dr. Rubeena Slamat

---

## [Decision Letter · Decision Letter 2]

9 Jan 2026

“Mother in the making”: Motherhood performativity of childless women in rural Pakistan

PONE-D-24-06562R2

Dear Dr. Slamat,

We’re pleased to inform you that your manuscript has been judged scientifically suitable for publication and will be formally accepted for publication once it meets all outstanding technical requirements.

Kind regards,

Saeed Ahmad, PhD

Academic Editor

PLOS One

Additional Editor Comments (optional):

Reviewers' comments:

Reviewer's Responses to Questions

**Comments to the Author**

Reviewer #2: All comments have been addressed

2. Is the manuscript technically sound, and do the data support the conclusions?

Reviewer #2: (No Response)

3. Has the statistical analysis been performed appropriately and rigorously?

Reviewer #2: (No Response)

4. Have the authors made all data underlying the findings in their manuscript fully available?

Reviewer #2: (No Response)

5. Is the manuscript presented in an intelligible fashion and written in standard English?

Reviewer #2: (No Response)

Reviewer #2: This revised manuscript presents a well-executed qualitative study examining motherhood performativity among childless women in rural Pakistan through the lens of Butler's performativity theory. The authors have demonstrated good faith in addressing previous reviewer comments and have made substantial improvements throughout.

Strengths:

The manuscript makes an important contribution by documenting the lived experiences of childless women in a pronatalist society where motherhood is socially obligatory. The theoretical framework effectively demonstrates how women navigate cultural expectations through performative acts - from treatment-seeking to displaying maternal qualities toward others' children - even in the absence of biological motherhood. The methodology is sound, with 15 in-depth interviews and 6 focus group discussions providing rich empirical data. The authors have appropriately acknowledged ethical considerations, limitations, and have added practical policy recommendations.

Recommendation:

This manuscript is ready for publication. It makes a valuable contribution to scholarship on motherhood, childlessness, and gender performativity in South Asian contexts, and effectively demonstrates how cultural discourses shape women's lives while revealing spaces for limited agency within constraining structures.

**Do you want your identity to be public for this peer review?** For information about this choice, including consent withdrawal, please see our Privacy Policy

Reviewer #2: No

---

## [Editor Report · Acceptance letter]

PONE-D-24-06562R2

PLOS One

Dear Dr. Slamat,

I'm pleased to inform you that your manuscript has been deemed suitable for publication in PLOS One. Congratulations! Your manuscript is now being handed over to our production team.

Kind regards,

on behalf of

Dr. Saeed Ahmad

Academic Editor

PLOS One